# Hydrogen Protection Boosts the Bioactivity of *Chrysanthemum morifolium* Extract in Preventing Palmitate-Induced Endothelial Dysfunction by Restoring MFN2 and Alleviating Oxidative Stress in HAEC Cells

**DOI:** 10.3390/antiox12051019

**Published:** 2023-04-28

**Authors:** Yilin Gao, Oumeng Song, Min Wang, Xin Guo, Guanfei Zhang, Xuyun Liu, Jiankang Liu, Lin Zhao

**Affiliations:** 1Center for Mitochondrial Biology and Medicine, The Key Laboratory of Biomedical Information Engineering of Ministry of Education, School of Life Science and Technology, Xi’an Jiaotong University, Xi’an 710049, China; gaoyilin39@stu.xjtu.edu.cn (Y.G.); oumeng@stu.xjtu.edu.cn (O.S.);; 2School of Health and Life Sciences, University of Health and Rehabilitation Sciences, Qingdao 266071, China

**Keywords:** *Chrysanthemum morifolium* extract, hydrogen, endothelial dysfunction, mitochondria, oxidative stress

## Abstract

As the most important natural antioxidants in plant extracts, polyphenols demonstrate versatile bioactivities and are susceptible to oxidation. The commonly used ultrasonic extraction often causes oxidation reactions involving the formation of free radicals. To minimize the oxidation effects during the ultrasonic extraction process, we designed a hydrogen (H_2_)-protected ultrasonic extraction method and used it in *Chrysanthemum morifolium* extraction. Hydrogen-protected extraction improved the total antioxidant capacity, 2,2-diphenyl-1-picrylhydrazyl (DPPH) radical scavenging activity, and polyphenol content of *Chrysanthemum morifolium* water extract (CME) compared with air and nitrogen (N_2_) conditions. We further investigated the protective effects and mechanisms of CME on palmitate (PA)-induced endothelial dysfunction in human aorta endothelial cells (HAECs). We found that hydrogen-protected CME (H_2_-CME) best-prevented impairment in nitric oxide (NO) production, endothelial NO synthase (eNOS) protein level, oxidative stress, and mitochondrial dysfunction. In addition, H_2_-CME prevented PA-induced endothelial dysfunction by restoring mitofusin-2 (MFN2) levels and maintaining redox balance.

## 1. Introduction

Plant extracts have a long history of application in medicines and meal supplements for their health benefits. *Chrysanthemum morifolium* originates in China and has been used as a “food and medicine dual-use” plant for over 2000 years [1]. *Chrysanthemum morifolium* tea is quite popular in daily life throughout East and Southeast Asia. The bioactive compounds isolated from *Chrysanthemum morifolium* include flavonoids, phenolic acids, anthraquinones, polysaccharides, and terpenoids [2]. According to previous investigations, the potential health benefits of *Chrysanthemum morifolium* may include, but are not limited to, antioxidant [3,4,5], anti-inflammatory [2,6,7], hypolipidemic effects [8], and anti-carcinogenesis activities [9,10], as well as cardiovascular protection [11,12]. Li, et al. [6] reported that *Chrysanthemum morifolium* extract (CME) inhibited lipopolysaccharide-induced increases in IL6 and IL1β mRNA levels, as well as H_2_O_2_-induced reactive oxygen species (ROS) production in cultured cells. Tian, et al. [13] demonstrated that CME protected against ethanol- and carbon tetrachloride-induced liver injury in vivo, while also inhibiting hepatocyte oxidative stress and apoptosis induced by acetaminophen in vitro through the activation of the nuclear factor erythroid-2-related factor 2(NRF2) signaling pathway.

Polyphenolic compounds, the most important natural antioxidants in plants, have many beneficial effects on human health and are mainly divided into flavonoids, phenolic acids, tannins, anthocyanins, and lignans [14,15,16]. Flavonoids and phenolic acids are abundantly contained in CME and are also their key active ingredients and nutrients [6,17]. Polyphenols are susceptible to being oxidized. Therefore, the extraction process largely affects the composition and content of polyphenols in plant extracts [15]. Ultrasound-assisted extraction (UAE) is one of the most promising techniques for modern plant extraction, which is simple and inexpensive, with a short extraction time, low solvent consumption, and high extraction rate compared with conventional extraction methods [18,19]. UAE ensures sufficient contact of the sample with the solvent and has been widely used to extract various plant samples. However, due to the cavitation phenomenon, free radicals are generated during UAE, which can reduce the biological activity of the extracts. The cavitation bubbles burst violently during the ultrasound, which causes high local temperature and pressure. H· radicals and OH· radicals will be generated when water and dissolved oxygen exist in the cavitation bubbles [20,21,22]. In this course, polyphenols react with free radicals, which reduces extraction efficiency. Ding, et al. [20] extracted rapeseed meals by continuously aerating nitrogen during UAE to increase polyphenol extraction yield. Due to the low number of related studies, developing more effective auxiliary extraction methods to increase the content and activity of easily oxidized compounds has become the focus of research. In this study, we designed a hydrogen-protected extraction method to minimize the oxidation effects during the UAE process. No study to date has examined hydrogen protection on plant extraction. Hydrogen is the smallest and lightest molecule and can diffuse rapidly through the cell membrane [23]. In the last decade, hydrogen medicine has received increasing attention for its biological activity as an emerging therapy [24,25,26,27,28,29,30,31,32].

The vascular endothelium is vital in maintaining multi-organ health and vascular dynamic homeostasis [33]. Endothelial dysfunction contributes to many cardiovascular and metabolic diseases, including atherosclerosis, hypertension, cardiomyopathy, and diabetes [34]. Endothelial dysfunction is characterized by impaired vasodilation, increased oxidative stress, inflammation, leukocyte adhesion, and endothelial-to-mesenchymal transition (EndoMT). The impairment of vasodilation is an important aspect of endothelial dysfunction [33]. Endothelial cells (ECs) control vasodilation by releasing endogenous vasodilator NO produced by eNOS. Endothelial-derived NO not only affects the tone of underlying vascular smooth muscle, but inhibits platelet aggregation and leukocyte adhesion [35]. When confronted with cardiovascular risk factors, ECs exhibit an overproduction of ROS, which subsequently triggers thrombogenic and pro-inflammatory pathways in the endothelium. The excessive superoxide reacts with NO to form peroxynitrite, consequently decreasing NO bioavailability and leading to endothelial dysfunction [36]. Previous studies have shown that CME attenuates oxidative damage in ECs [37,38], reduces oxidative stress-induced ECs adhesion [39], and has endothelium-dependent vasodilatory effects [40,41]. However, the mechanisms by which CME prevents endothelial dysfunction remain unclear. The objective of the present study was to conduct a comparative analysis of the protective effects of CME prepared under air, nitrogen, and hydrogen conditions against PA-induced endothelial dysfunction in vitro. Additionally, we aimed to investigate the underlying mechanism of CME’s endothelial protective effects with a focus on the regulation of redox balance and mitochondrial homeostasis.

A healthy mitochondrial network is vital for cardiovascular and endothelial function. Mitochondria regulate essential cellular processes, including metabolism, cell death, ATP (adenosine triphosphate) generation, and immune responses [42]. The regulation of these processes is usually associated with mitochondrial dynamics. Mitochondria are constantly undergoing fission or fusion in response to changes in the cellular environment and physiological demands. This is a dynamic equilibrium process. Disruption of mitochondrial dynamics leads to mitochondrial dysfunction and human diseases. Mitochondrial fusion promotes cellular oxidative phosphorylation and contributes to damage repair of mitochondrial DNA [43]. Mitofusin 1 (MFN1) and MFN2 play a crucial role in the tethering of adjacent mitochondria and executing the fusion of the outer mitochondrial membrane [44]. MFN2 also performs non-fusion roles, including modulating endoplasmic reticulum (ER)-mitochondria tethering, mediating mitophagy, and apoptosis [45]. Depletion of MFN2 leads to ER stress. Research has indicated that MFN2 exerts barrier stabilization and anti-inflammatory effects in endothelium by stabilizing cell–cell adherens junctions [46]. Various stressors, such as oxidative stress, hypoxia, and high glucose, have been demonstrated to induce mitochondrial fission in ECs. Disruption of mitochondrial dynamics mediates pathological changes in the endothelium, including endothelial inflammation, reduced microvasculature, impaired endothelium-dependent relaxation, and defects in angiogenesis [47,48,49,50]. However, it is unclear whether CME can regulate mitochondrial homeostasis in the endothelium. In this study, we demonstrated that CME reduced mitochondrial fragmentation and protected mitochondrial function. Additionally, CME exerted endothelial protective effects via restoring MFN2.

## 2. Materials and Methods

### 2.1. Preparation of CME

Dried flower heads of *Chrysanthemum morifolium* (CM) were purchased from Hangzhou Tea Co., Ltd. (Hangzhou, China). CM was ground in liquid nitrogen and sieved to obtain a fine powder. Hydrogen or nitrogen gas was injected into the water for 15 min before mixing with the CM powder (liquid-to-solid ratio of 30 mL/g). CM water extract was prepared using UAE at room temperature with 200 W for 20 min. The extract was filtered through a 0.45 µm filter to remove sample residue and lyophilized to remove water. For cell culture, hydrogen or nitrogen gas was injected into the water for 15 min, and then the lyophilized extract was dissolved in water and filtered through a 0.22 µm filter. The extraction process is shown in Figure 1a. The air group did not perform any aeration treatment during the extraction process.

### 2.2. Total Antioxidant Capacity Assay

Total antioxidant capacity (T-AOC) was measured by the T-AOC assay kit (BC1315, Solarbio, Beijing, China). The capacity of tripyridyltriazine-Fe^3+^ (TPTZ-Fe^3+^) reduction to tripyridyltriazine-Fe^2+^ (TPTZ-Fe^2+^) under acidic conditions reflects the total antioxidant capacity of the sample. The absorbance was measured at the wavelength of 593 nm by a microplate reader (Thermo Fisher Scientific, Waltham, MA, USA).

### 2.3. DPPH Radical Scavenging Activities Assay

The DPPH radical scavenging activities of CME were measured by a DPPH assay kit (BC4750, Solarbio) at nine concentrations ranging from 0.03125 mg/mL to 10 mg/mL. CME samples were mixed with DPPH solution and incubated for 10 min at 37 °C. The absorbance was measured at the wavelength of 515 nm using a microplate reader (Thermo Fisher Scientific). The IC50 values (mg/mL) of CME were calculated based on a logarithmic regression curve using GraphPad Prism 9.0 software (RRID: SCR_000306, La Jolla, CA, USA).

### 2.4. Liquid Chromatograph Mass Spectrometer (LC-MS) Analysis of Chlorogenic Acid (CA) and Luteolin (LU) Concentration

LC-MS analysis was performed by a Thermo-Fisher TSQ Quantis. CA and LU standard compounds were purchased from YuanYe Co., Ltd. (Shanghai, China). Dried CME samples were dissolved in methanol and filtered through 0.22 µm filters. A 2uL sample was injected for analysis. ACQUITY UPLC BEH C18 column (2.1 mm × 50 mm, 1.7 µm) was used for chromatographic separation. The system was maintained at 45 °C, while the mobile phase had a flow rate of 0.2 mL/min and consisted of phase A (0.1% formic acid in water) and phase B (methanol). Full scan mass spectra were obtained in the range of 100–1000 *m*/*z*. The electron spray ionization conditions were as follows: capillary voltage: 2.8 kV, capillary temperature: 325 °C, nitrogen sheath gas flow: 35 arbitrary units, auxiliary gas flow: 10 arbitrary units, negative mode.

### 2.5. Cell Culture and Treatment

Human aorta endothelial cells (HAECs) were obtained from BioLeaf Biotech Co., Ltd. (Shanghai, China) and cultured in DMEM medium (10-013-CVRC, Corning, New York, NY, USA) with 10% fetal bovine serum (FBS) (04-001-1ACS, Biological Industries, Kibbutz Beit Haemek, Israel), 100 U/mL penicillin G (P3032, Sigma, Darmstadt, Germany), 100 μg/mL streptomycin sulfate (11860038, Thermo Fisher Scientific), 10 μg/mL heparin (H8060, Solarbio). The cells were maintained in a humidified incubator with 5% CO_2_ at 37 °C and used between passage 2 and 6. At 75–85% confluence, cells were seeded into the cell-cultured plate and pretreated with different types of CME (100 μg/mL) for 24 h, followed by PA (500 µM) or 10% bovine serum albumin (BSA, solvent control) for 24 h.

### 2.6. PA Stock Solution Preparation

Sodium PA (P9767, Sigma) was dissolved in NaOH-ethanol solution at 80 °C for 10 min to prepare 200 mM stock solution and then diluted with DMEM medium containing 10% fatty acid-free BSA (0219989980, MP Biomedicals, Irvine, CA, USA) at the ratio of 1:19 to make a final stock solution of 5 mM PA.

### 2.7. Cell Viability Measurement

Cell viability was measured by the 3-[4,5-dimethylthiazol-2-yl]-2,5-diphenyl tetrazolium bromide (MTT) (M2128, Sigma) assay in 96-well culture plates. After treatment, cells were incubated with MTT (0.5 mg/mL) dissolved in the FBS-free DMEM medium for 4 h at 37 °C. Then, the medium was removed, and 150 μL dimethyl sulfoxide (DMSO) was added to each well to dissolve the formazan. The absorbance was measured at the wavelength of 490 nm using a microplate reader (Thermo Fisher Scientific).

### 2.8. NO Production Measurement

The generated NO was measured by a NO-specific fluorescent probe 4,5-diaminofluorescein (DAF-FMDA) (S0019, Beyotime, Shanghai, China). Intracellular esterases metabolize DAF-FMDA to a non-fluorescent molecule DAF-FM, which reacts with NO to the strong-fluorescence product (Ex: 495 nm, Em: 515 nm). After treatment, cells were incubated with DAF-FMDA (5 µM) for 30 min at 37 °C in the dark and washed three times with PBS. Cells were observed under an inverted fluorescent microscope. Fluorescence intensity was analyzed using the Image J software (National Institutes of Health, RRID: SCR_003070, Bethesda, MD, USA).

### 2.9. Protein Extraction and Western Blot

For whole cell lysates, HAECs (2–3 × 10^6^ cells) were lysed with 100 μL lysis buffer (P10013, Beyotime) supplemented with 2% protease inhibitors and 2% phosphatase inhibitors (P1050, Beyotime) for 10 min on ice. The lysates were centrifuged at 13,000× *g* for 10 min at 4 °C. Protein concentration of the collected supernatants was measured by BCA Protein Assay Kit (23225, Thermo Fisher Scientific). Nuclear and cytoplasmic lysates were prepared by Nuclear and Cytoplasmic Protein Extraction Kit (P0027, Beyotime). Equal amounts of protein samples (10 μg) were separated by SDS-PAGE, transferred to a nitrocellulose membrane, blocked with 5% nonfat milk in TBST buffer at room temperature for 1–2 h, and analyzed by immunoblotting. Membranes were incubated with specific primary antibodies (for details, see Appendix A) overnight at 4 °C and washed three times in TBST buffer before incubation in HRP-conjugated secondary antibody (111-035-003/115-035-003, Jackson Immunoresearch, West Grove, PA, USA). Membranes were washed three times again in TBST buffer before visualization with ECL substrate (1705061, Bio-Rad, Hercules, CA, USA). Images were acquired with the Clinx Chemi Analysis system (Clinx, Shanghai, China) and quantified with Image J software (National Institutes of Health, RRID: SCR_003070).

### 2.10. Quantitative Real Time-PCR (qPCR)

Cells were collected with Trizol reagent (9109, Takara, Tokyo, Japan), and total RNA was extracted according to the manufacturer’s protocol. RNA was used to synthesize cDNA using the cDNA Synthesis SuperMix Kit (AU341-02, Trans, Beijing, China). cDNA was performed for qPCR analysis using SYBR Green qPCR SuperMix (AU601-02, TRANS) and specific primers (for primer sequences, see Appendix A). *β-ACTIN* was used as an endogenous control gene. Fold change was evaluated by the 2^−∆∆CT^ method for comparing gene expression under different conditions.

### 2.11. Intracellular ROS Measurement

For intracellular ROS detection, we used two types of ROS-specific fluorescent probes: Dihydroethidium (DHE) (S0063, Beyotime) and 2′,7′-Dichlorofluorescin diacetate (DCF-DA) (D6883, Sigma).

(1) DHE-based ROS measurement: DHE is dehydrogenated by intracellular superoxide anions (O_2_^−^) to produce ethidium. Ethidium binds to RNA or DNA to produce red fluorescence (Ex: 300 nm, Em: 610 nm). After treatment, cells were fixed in 4% paraformaldehyde for 10 min at room temperature and incubated with DHE (1 µM) in PBS for 30 min at 37 °C in the dark and then washed three times with PBS. Cells were observed under an inverted fluorescent microscope. Fluorescence intensity was analyzed using the Image J software (National Institutes of Health, RRID: SCR_003070);

(2) DCF-DA-based ROS measurement: DCF-DA is metabolized to a non-fluorescent molecule by intracellular esterases and then oxidized by H_2_O_2_ to the strong-fluorescence product DCF (Ex: 504 nm, Em: 529 nm). After treatment, cells were incubated with DCF-DA (10 µM) in the FBS-free DMEM medium for 30 min at 37 °C in the dark, then washed three times with PBS. Next, cells were lysed and centrifuged at 13,000× *g* for 10 min at 4 °C. Finally, the supernatant was collected, and the fluorescence intensity was measured using a microplate fluorometer (Thermo Fisher Scientific).

### 2.12. Mitochondrial Superoxide Measurement

Mitochondrial superoxide was measured by a Mitochondrial-superoxide-specific fluorescent probe MitoSOX Red (M36008, Invitrogen, Carlsbad, CA, USA). MitoSOX can be oxidized by superoxide in live cells to produce red fluorescence (Ex: 396 nm, Em: 610 nm). Cells were incubated with MitoSOX (5 µM) in FBS-free DMEM medium for 15 min at 37 °C after treatment, then washed three times with PBS and observed under an inverted fluorescent microscope. Fluorescence intensity was analyzed by Image J software (National Institutes of Health; RRID: SCR_003070).

### 2.13. Mitochondrial Morphology Measurement

Mitochondrial morphology was measured by a mitochondrial-specific fluorescent probe MitoTracker Red FM (M22425, Invitrogen). MitoTracker binds to mitochondria in living cells and produces red fluorescence (Ex: 581 nm, Em: 644 nm). Cells were incubated with MitoTracker (0.3 µM) in FBS-free DMEM medium for 10 min at 37 °C after treatment, then washed three times with PBS and observed under an inverted fluorescent microscope. Extracting the skeleton after binary processing of the images. Fluorescence intensity and mean mitochondrial length were analyzed by Image J software (National Institutes of Health, RRID: SCR_003070).

### 2.14. Mitochondrial DNA (mtDNA) Copy Number Measurement

After treatment, cells were collected, and then genomic DNA was extracted and purified using a QIAamp DNA Mini Kit (51304, Qiagen, Duesseldorf, Germany). qPCR was used to measure the mtDNA copy number using *D-LOOP* as the target gene and *18S rRNA* as an endogenous control gene (for primer sequences, see Appendix A). Fold change was evaluated by the 2^−∆∆CT^ method for comparing *D-LOOP* gene expression under different conditions to represent mtDNA copy number.

### 2.15. Oxygen Consumption Rate (OCR) Measurement

Cells were seeded at 20,000 cells per well on Seahorse XF24 culture plates (Seahorse Bioscience, Billerica, MA, USA). After treatment, cells were incubated in assay media with 10 mM glucose, 1 mM pyruvate, and 2 mM glutamine in a non-CO_2_ incubator at 37 °C for 1 h. Oxygen consumption was measured using an XF24 Extracellular Flux Analyzer (Seahorse Bioscience, Billerica, MA, USA) according to the manufacturer’s protocol with inhibitors (0.75 µM oligomycin, 0.75 µM carbonyl cyanide 4-(trifluoromethoxy)phenylhydrazone (FCCP), 2 µM antimycin A). Basal respiration, ATP production, maximal respiration, and spare capacity were calculated from measured oxygen consumption.

### 2.16. siRNA Transfection for MFN2 and SOD2 Knockdown

The siRNA oligo was purchased from Genepharma Co., Ltd. (Shanghai, China) (for sequences, see Appendix A). When cells at 30% confluence, targeting siRNA or non-targeting control siRNA was diluted in Opti-MEM Reduced Serum Medium (11058021, Gibco, Grand Island, NY, USA) and mixed with Lipofectamine RNA-iMAX Transfection Reagent (13778150, Invitrogen), then added to the culture medium for 6 h.

### 2.17. Statistical Analysis

GraphPad Prism 9.0 software (RRID: SCR_000306) was used to generate graphs and perform statistical analysis. All data are expressed as mean ± SEM. Normal distribution was assessed by the Shapiro–Wilk test. The homogeneity of variances was assessed by the Brown–Forsythe test. For normally distributed data, one-way ANOVA followed by post hoc Tukey or Sidak test was performed for analyses if there was no significant variance inhomogeneity. For data that deviate from a normal distribution, the Kruskal–Wallis test, followed by the post hoc Dunn test, was performed for analyses. *p* < 0.05 was considered to be statistically significant for all comparisons.

## 3. Results

### 3.1. Hydrogen-Protected Extraction Increased the Content of Polyphenolic Compounds in CME and Enhanced the Antioxidant Capacity of CME

To enhance the biological activity of CME, we designed a hydrogen-protected extraction method based on UAE. Figure 1a illustrates the hydrogen-protected extraction process. The specific method has been described in Section 2.1. First, we wanted to verify whether the antioxidant activity of CME was affected by the different extraction methods. We compared the antioxidant activity of different extraction conditions of CME using two antioxidant assays (Figure 1b,c). The results suggested that hydrogen protection and nitrogen protection enhanced the total antioxidant capacity and DPPH radical scavenging activity of CME compared to the air conditions, and the H_2_ group had better antioxidant capacity than the N_2_ group.

Luteolin (LU) and chlorogenic acid (CA) are abundant in CME; they are flavonoids and phenolic acids, respectively, and both are easily oxidized polyphenolic compounds. Compared with the air group, hydrogen protection and nitrogen protection could significantly increased the luteolin content in the CME, and hydrogen protection was more effective than nitrogen protection (Figure 1d). Moreover, hydrogen protection could increase the content of chlorogenic acid in the CME, while nitrogen protection could not (Figure 1e).

Since the follow-up experiments were performed in HAECs, we tested the toxicity of CME in different extraction conditions in HAECs. 1000 μg/mL or less CME did not affect cell viability and morphology (Appendix A).

### 3.2. Hydrogen-Protected Extraction Enhanced the Capacity of CME to Attenuate PA-Induced Endothelial Dysfunction in HAEC Cells

Increased circulating levels of fatty acids, including PA, are an important trigger of endothelial dysfunction. Our previous study demonstrated that 500 μM PA treatment induced endothelial dysfunction in HAECs [51]. Although CME has been reported to possess antioxidant and anti-inflammatory properties, it is unclear whether CME treatment can reverse PA-induced endothelial dysfunction in vitro. As shown in Figure 2a,b, pretreatment with 100 μg/mL CME for 24 h prevented the decrease in cell viability and protected the morphological damage induced by PA. The H_2_ condition was the most effective in protecting cell viability.

ECs control vasodilation by releasing endogenous vasodilator NO produced by eNOS. Phosphorylation at Ser1177 activates eNOS [33]. We found that CME reversed the decrease in intracellular NO levels induced by PA. Notably, the NO levels in the H_2_ group were significantly higher than in the air and N_2_ groups (Figure 2c). Consistent with the changes in NO levels, we found decreased protein levels of eNOS and p-eNOS^Ser1177^ in HAECs challenged by PA (Figure 2d). Interestingly, only H_2_-CME pretreatment reversed the eNOS level, H_2_-CME and N_2_-CME reversed the p-eNOS^Ser1177^ level, and all three CME reversed the p-eNOS/eNOS ratio.

In response to injury, ECs are activated and produce inflammatory factors. These factors recruit monocytes and neutrophils, which ultimately lead to vascular inflammation [33]. Treatment with PA led to significant up-regulation of inflammatory factors IL6 and MMP1 at the mRNA levels (Figure 2e,f). CME pretreatment significantly inhibited the upregulation of IL6 and MMP1 gene expression, and the H_2_ conditions had the most pronounced effects.

Metabolic disarrangements and inflammation in ECs can impose high demands on the ER protein-folding machinery, thereby triggering ER stress. In turn, ER stress induces inflammation and oxidative stress [52]. We found that PA caused ER stress, while CME inhibited PA-induced upregulation of gene expression of ER stress markers CHOP, GRP78, GRP94, and XBP1(Figure 2g–j). Furthermore, from the mRNA levels of CHOP and GRP78, the inhibition effects of ER stress by CME under H_2_ conditions were more effective than air and N_2_ conditions.

Together, these results suggested that hydrogen-protected extraction enhanced the bioactivity of CME, whether by protecting cell viability, increasing NO levels, anti-inflammation, or inhibiting ER stress.

### 3.3. Hydrogen-Protected Extraction Enhanced the Capacity of CME to Attenuate PA-Induced Oxidative Stress in HAEC Cells

Oxidative stress is a trigger for endothelial dysfunction. We next investigated whether PA-induced oxidative stress in HAECs. We used two specific fluorescent probes, DHE and DCF-DA, to detect intracellular ROS levels. Both results indicated that PA significantly upregulated intracellular ROS levels, while CME treatment inhibited the PA-induced increase in ROS levels (Figure 3a,b). Importantly, the H_2_ group had a lower ROS level than the air group. We next assessed the total antioxidant capacity. As expected, PA reduced the total antioxidant capacity of HAECs. Only CME treatment with H_2_ and N_2_ conditions could restore the total antioxidant capacity, while air conditions could not (Figure 3c).

Considering the primary contribution of phase II enzymes to maintaining cellular redox homeostasis, we next analyzed the protein levels of endogenous phase II enzymes. As shown in Figure 3d, kelch-like ECH-associated protein 1 (KEAP1) significantly decreased after PA treatment, but protein and mRNA levels of NRF2 remained unchanged (Appendix A). We further analyzed the nucleus translocation of NRF2, but no significant changes were observed (Appendix A). No significant changes were observed in the catalase protein levels (Figure 3d). NAD(P)H: quinone oxidoreductase 1 (NQO1) protein levels increased after PA treatment (Figure 3d). Superoxide dismutase (SOD) is the major antioxidant defense system against superoxide [53]. In mammals, it consists of three isozymes: Cu/ZnSOD (SOD1) in the cytoplasm, MnSOD (SOD2) in the mitochondria, and extracellular Cu/ZnSOD (SOD3) [54]. Interestingly, treatment with PA led to the downregulation of SOD1 and significant up-regulation of SOD2 at the protein levels, while SOD2 protein levels decreased after CME treatment for all three conditions (Figure 3d). No significant changes were observed in the mRNA levels of SOD2 (Appendix A). This phenomenon will be investigated and discussed in depth later.

### 3.4. Hydrogen-Protected Extraction Enhanced the Capacity of CME to Attenuate PA-Induced Mitochondrial Fragmentation and Dysfunction in HAEC Cells

Due to enhanced oxidative stress, the endogenous antioxidant enzyme SOD2 is compensatorily upregulated in PA-induced HAECs. Therefore, we measured the mitochondrial superoxide levels. As expected, mitochondrial superoxide levels increased significantly after PA treatment. CME pretreatment inhibited the increase of mitochondrial superoxide, and the effect of the H_2_ group was better than the air group (Figure 4a).

Disruption of mitochondrial dynamics leads to mitochondrial dysfunction and human diseases [43]. We observed that PA caused significant mitochondrial fragmentation, as evidenced by a decrease in mean mitochondrial branch length (Figure 4b). CME pretreatment reduced the extent of PA-induced mitochondrial fragmentation. No significant changes in mitochondrial content were observed (Figure 4b). As significant changes in mitochondrial morphology were observed, the levels of fission and fusion proteins were next investigated (Figure 4c). PA treatment significantly decreased the level of mitochondrial fission protein dynamin-related protein 1 (DRP1), and CME treatment in H_2_ and N_2_ conditions reversed the DRP1 protein level. Treatment with PA also led to the downregulation of mitochondrial fusion proteins MFN1 and MFN2. Only H_2_ conditions could increase MFN1 and MFN2 protein levels after CME pretreatment. No significant changes in fusion protein optic atrophy protein 1 (OPA1) levels were observed.

To investigate the effects of CME on mitochondrial mass and energy metabolism, we measured mitochondrial DNA (mtDNA) copy number and mitochondrial oxygen consumption rate (OCR). As shown in Figure 4d, CME significantly reversed the decrease in mtDNA copy number induced by PA. The results of OCR indicated that mitochondrial basal respiration, ATP production, and maximal respiration showed a decreasing trend after PA treatment, which was prevented by CME pretreatment in H_2_ conditions (Figure 4e). However, this part of the data was not statistically significant. The mitochondrial electron transfer chain (ETC) comprises five enzyme complexes. Western blot results of the five complex subunits showed decreased levels of NDUFS3 (complex I), SDHB (complex II), and UQCRC1 (complex III) in HAECs stimulated by PA. CME pretreatment in H_2_ and N_2_ conditions reversed NDUFS3 levels (Figure 4f). We did not observe significant changes in MTCO1 (complex IV) and ATP5A (complex V) protein levels.

### 3.5. MFN2 Mediated H_2_-CME’s Prevention of PA-Induced Decrease of NO Production and Increase of ROS Generation in HAEC Cells

Among the PA-induced downregulation of mitochondrial fission and fusion proteins, the most pronounced change was in the MFN2 protein level, which decreased by about 75% (Figure 4c). MFN2 has been reported as a stabilizer of the vascular endothelial barrier [46]. In this study, we evaluated the role of MFN2 absence in PA-induced endothelial dysfunction. Knockdown of MFN2 caused mitochondrial fragmentation (Figure 5a), similar to the mitochondrial morphological changes induced by PA treatment (Figure 4b). We found that MFN2 knockdown reduced NO levels in HAECs and inhibited the protective effect of H_2_-CME in the PA-induced decrease in NO levels (Figure 5b). Knockdown of MFN2 also led to significant downregulation of eNOS and p-eNOS^Ser1177^ at the protein levels (Figure 5c). In MFN2 knockdown HAECs, H_2_-CME failed to protect cells from PA-induced decrease in eNOS and p-eNOS^Ser1177^ protein levels. The knockdown of MFN2 had no significant effect on the SOD2 protein levels. Furthermore, MFN2 knockdown upregulated intracellular ROS levels (Figure 5d). In the MFN2 knockdown condition, H_2_-CME failed to protect HAECs from PA-induced overproduction of ROS. These findings indicated a protective role of MFN2 in maintaining endothelial function. MFN2 partially mediated the H_2_-CME’s prevention of PA-induced NO decrease and ROS generation in HAECs.

### 3.6. Oxidative Stress Mediated PA-Induced Endothelial Dysfunction in HAEC Cells

We have shown that PA caused oxidative stress in HAECs (Figure 3a,b). Given that oxidative stress is an important trigger of endothelial dysfunction, we wanted to assess whether PA led to endothelial dysfunction via inducing oxidative stress. First, we used tert-Butyl hydroperoxide (TBHP) and Antimycin A to induce oxidative stress. TBHP treatment is usually used as a positive control for cellular oxidative stress. Antimycin A is an ETC inhibitor that inhibits the electron transfer from complex III to ubiquinone. The use of antimycin increases mitochondrial superoxide generation and depletes the mitochondrial membrane potential [55]. TBHP or Antimycin A treatment successfully induced an increase in ROS levels (Figure 6a). Endothelial dysfunction was observed in PA-treated HAECs, including impaired total antioxidant capacity, cell viability, NO levels, and eNOS, MFN2 and UQCRC1 protein levels, as well as significant up-regulation of IL6, MMP1, and CHOP mRNA levels (Figure 2b–g, Figure 3c and Figure 4c,f). Similarly, these phenomena were observed in TBHP- and antimycin-treated HAECs (Figure 6b–h). Therefore, we considered that PA may cause endothelial dysfunction via oxidative stress.

We next used ROS inhibitor N-acetylcysteine (NAC) to inhibit PA-induced oxidative stress. NAC and PA co-treatment successfully reduced ROS levels (Figure 6a). In addition, co-treatment with PA and NAC reduced endothelial dysfunction, including restoration of total antioxidant capacity, cell viability, NO levels, and eNOS, DRP1, MFN2, NDUFS3, and UQCRC1 protein levels, as well as inhibition of IL6 and CHOP mRNA levels (Figure 6b–h). These results suggested that PA-induced oxidative stress caused endothelial dysfunction in HAECs.

As shown in Figure 3d, PA significantly upregulated the SOD2 protein levels, while CME treatment decreased the SOD2 protein levels. We considered that PA treatment induced a large amount of mitochondrial superoxide production, and HAECs scavenged the excess mitochondrial superoxide by upregulating SOD2 as an adaptive response. CME treatment inhibited the production of mitochondrial superoxide, leading to a decrease in the SOD2 protein level. As shown in Figure 6e, TBHP and antimycin treatments also led to the upregulation of SOD2 at the protein levels. Compared to the PA group, NAC treatment decreased the SOD2 protein levels. These results confirmed our inference.

### 3.7. H_2_-CME Prevented PA-Induced Decrease in NO Level by Maintaining Redox Balance in HAEC Cells

Since CME inhibited the PA-induced increase in ROS levels (Figure 3a,b and Figure 4a) and SOD2 protein levels (Figure 3d), we considered that CME exerted endothelial protective effects by inhibiting oxidative stress.

We next disrupted redox homeostasis in HACEs by knocking down SOD2 to assess whether CME plays an endothelial protective role, mainly by maintaining redox homeostasis or inhibiting oxidative stress. Knockdown of SOD2 successfully induced an increase in mitochondrial superoxide and total cellular ROS level (Figure 7a,b). However, in the SOD2 knockdown HAECs, H_2_-CME failed to reduce the PA-induced increase in mitochondrial superoxide and total ROS level. In addition, the protective effects of H_2_-CME on PA-induced decrease of NO levels were significantly weakened by SOD2 knockdown (Figure 7c). Furthermore, H_2_-CME failed to recover the PA-induced decrease in protein levels of eNOS, p-eNOS^Ser1177^ and MFN2 after SOD2 knockdown (Figure 7d). These results suggested that H_2_-CME prevented PA-induced endothelial dysfunction by maintaining redox balance.

## 4. Discussion

Polyphenolic compounds are the most important natural antioxidants in plant extracts. Polyphenols have potent antioxidant activity and are susceptible to being oxidized. The extraction process largely affects the composition and content of polyphenols in plant extracts. To minimize the oxidation effects and increase the content of polyphenols during the extraction process, we designed a hydrogen-protected extraction method based on UAE and used it for CM extraction. No study to date has examined hydrogen protection on plant extraction. We demonstrated that different extraction methods exhibited different antioxidant efficacies. Hydrogen-protected extraction enhanced the total antioxidant capacity and DPPH radical scavenging activity of CME and increased the content of luteolin and chlorogenic acid.

We compared the protective effects of CME under air, N_2_, and H_2_ conditions against PA-induced endothelial dysfunction in HAECs. Under H_2_ conditions, CME better protected against PA-induced decreases in cell viability, NO, and eNOS levels. Moreover, H_2_-CME had more pronounced effects in anti-inflammation and inhibition of ER stress. In PA-induced oxidative stress, the H_2_-CME condition had higher total antioxidant capacity, lower total ROS, and mitochondrial superoxide levels. To better investigate the protective effects and mechanism of action of H_2_-CME, further in vivo studies need to be checked in the future.

Mitochondria constantly undergo fission and fusion in response to changes in the cellular environment and physiological demands. This is a dynamic equilibrium process. Disruption of mitochondrial dynamics mediates pathological changes in the endothelium, including endothelial inflammation, reduced microvasculature, impaired endothelium-dependent relaxation, and defects in angiogenesis [47,48,49,50]. We observed that PA caused significant fragmentation in HAECs, and both the fission protein DRP1 and the fusion proteins MFN1 and MFN2 were significantly downregulated. We considered that mitochondrial dynamics were disrupted and that both fission and fusion processes were inhibited. Mitochondrial fusion was more severely impaired, leading to increased fragmentation. CME in all three conditions ameliorated mitochondrial fragmentation. H_2_-CME pretreatment could increase DRP1, MFN1, and MFN2 protein levels. It has been reported that MFN2 exerts barrier stabilization and anti-inflammatory effects in endothelium by stabilizing cell–cell adherens junctions [46]. Our research indicated that the knockdown of MFN2 led to the downregulation of NO, eNOS, and p-eNOS^Ser1177^ protein levels. In MFN2 knockdown HAECs, H_2_-CME failed to protect HAECs from PA-induced decrease in NO, eNOS, and p-eNOS^Ser1177^ protein levels. These findings indicated an essential role of MFN2 in endothelial cells, suggesting that a careful investigation is warranted for the use of MFN2 as a therapeutic target in the treatment of endothelial dysfunction.

SOD catalyzes the conversion of superoxide into oxygen and hydrogen peroxide [53]. Moreover, SOD plays a critical role in inhibiting the oxidative inactivation of NO, thereby preventing the formation of peroxynitrite and endothelial dysfunction [56]. Mammals have three isoforms of SOD (SOD1 in the cytoplasm, SOD2 in the mitochondria, and extracellular SOD3). Each is a product of distinct genes and subcellular localization but catalyzes the same reaction [54]. In this study, treatment with PA led to the downregulation of SOD1 and significant up-regulation of SOD2 at the protein level, while the SOD2 protein level decreased after CME treatment. We considered that PA treatment induced a large amount of mitochondrial superoxide production, and HAECs scavenged the excess mitochondrial superoxide by upregulating SOD2. CME treatment inhibited the production of ROS, leading to a decrease in SOD2 levels. We used TBHP and antimycin A to induce oxidative stress and NAC to inhibit ROS production. TBHP and antimycin treatments also led to the upregulation of SOD2 at the protein level. NAC treatment reduced the PA-induced upregulation of the SOD2 protein level. These results confirmed that the upregulation of SOD2 in HAECs was due to the overproduction of ROS.

We also evaluated the role of oxidative stress in PA-induced endothelial dysfunction. TBHP and antimycin treatment caused endothelial dysfunction similar to PA treatment, including impaired total antioxidant capacity, cell viability, NO levels, eNOS protein levels, inflammation, and ER stress. NAC treatment inhibited the decrease in total antioxidant capacity, cell viability, NO levels, eNOS protein levels induced by PA, and the upregulation of IL6 and CHOP gene expression. NAC treatment also inhibited the PA-induced decrease in mitochondrial protein levels of DRP1, MFN2, NDUFS3, and UQCRC1. These results confirmed that PA-induced oxidative stress led to endothelial and mitochondrial dysfunction in HAECs. Lastly, we disrupted redox homeostasis in HACEs by knocking down SOD2. The absence of SOD2 canceled the protective effects of H_2_-CME at mitochondrial superoxide level, total ROS level, NO levels, and eNOS and MFN2 protein levels. These results suggested that H_2_-CME prevented PA-induced endothelial dysfunction by maintaining redox homeostasis.

Our research, for the first time, demonstrated that hydrogen-protected extraction best improved the antioxidant capacity and polyphenol content of CME, as well as the protective efficacy of CME in PA-induced endothelial dysfunction, mitochondrial impairment, and oxidative stress in HAECs. Moreover, we found that H_2_-CME prevented PA-induced endothelial injury mainly by maintaining mitochondrial homeostasis and redox balance. Our research not only provides feasibility for the use of hydrogen protection in plant extraction but also highlights the significance of H_2_-CME in preventing endothelial dysfunction.

## Figures and Tables

**Figure 1 antioxidants-12-01019-f001:**
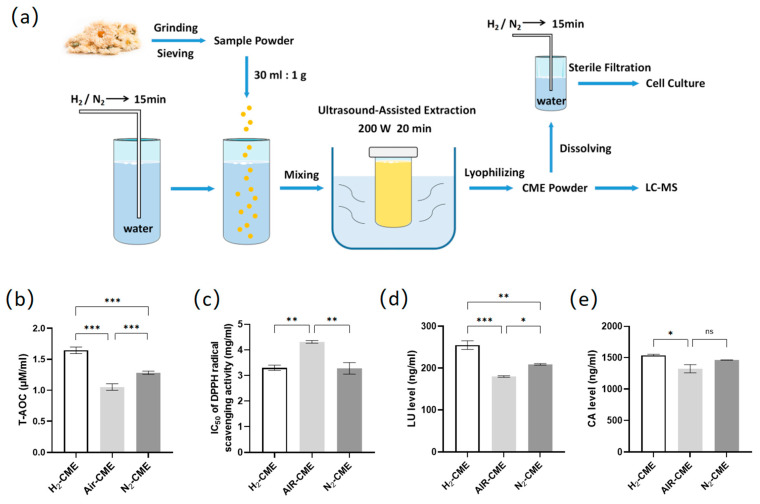
Hydrogen-protected extraction increased the content of polyphenolic compounds in CME and enhanced the antioxidant capacity of CME. (**a**) Extraction process of CME; (**b**) total antioxidant capacity of 1 mg/mL CME. n = 4; (**c**) IC50 of DPPH radical scavenging activities from CME. n = 3. (**d**,**e**) Luteolin (**d**) and chlorogenic acid (**e**) content of CME analyzed by LC-MS. n = 3. Values are mean ± SEM, * *p* < 0.05, ** *p* < 0.01, *** *p* < 0.001, ns: not significant.

**Figure 2 antioxidants-12-01019-f002:**
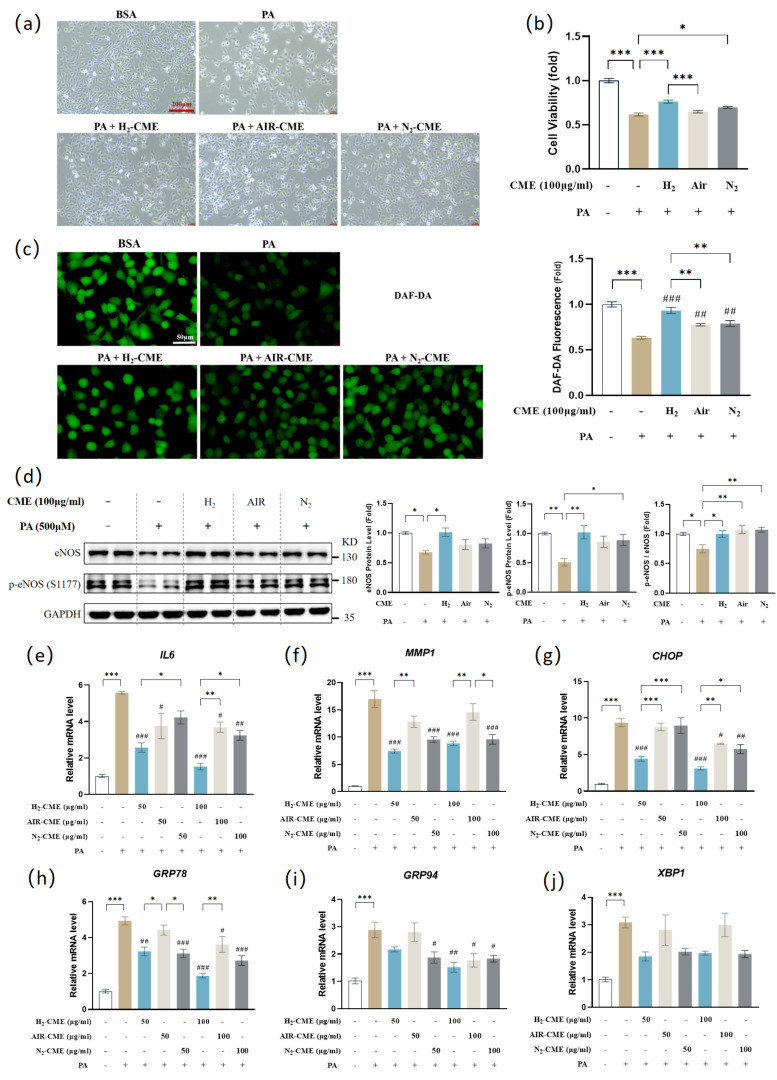
Hydrogen-protected extraction enhanced the capacity of CME to attenuate PA-induced endothelial dysfunction in HAEC cells. HAECs were pretreated with 50 or 100 μg/mL CME for 24 h, followed by 500 μM PA or BSA treatment for another 24 h. (**a**) Observation of cell morphology using an inverted microscope. Magnification: ×10. Scale bar = 200 μm; (**b**) Cell viability was measured by MTT assay. n = 12; (**c**) Intracellular NO level was visualized by DAF-FMDA fluorescent probe and quantified. Magnification: ×40. Scale bar = 50 μm; (**d**) Western blot analysis and densitometric quantification of the protein levels of eNOS, p-eNOS(S1177), and GAPDH. n = 4; (**e**–**j**) mRNA levels of IL6 (**e**), MMP1 (**f**), CHOP (**g**), GRP78 (**h**), GRP97 (**i**) and XBP1 (**j**) was measured by qPCR. n = 4. Values are mean ± SEM. * *p* < 0.05, ** *p* < 0.01, and *** *p* < 0.001 matched indicated group. ^#^
*p* < 0.05, ^##^
*p* < 0.01, and ^###^
*p* < 0.001 matched PA group.

**Figure 3 antioxidants-12-01019-f003:**
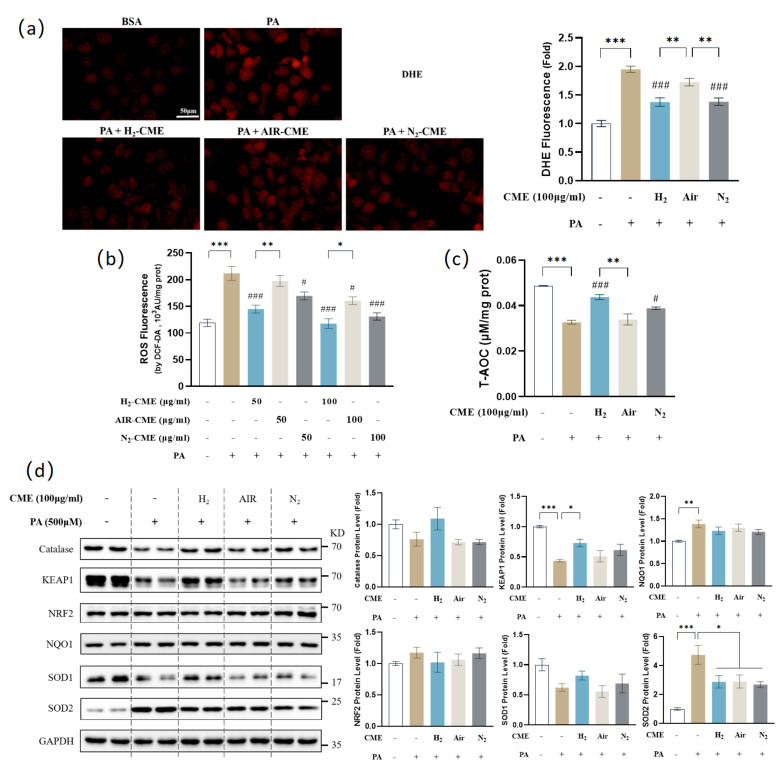
Hydrogen-protected extraction enhanced the capacity of CME to attenuate PA-induced oxidative stress in HAEC cells. HAECs were pretreated with 50 or 100 μg/mL CME for 24 h, followed by 500 μM PA or BSA treatment for another 24 h. (**a**) Intracellular ROS level was visualized by DHE fluorescent probe and quantified. Magnification: ×40. Scale bar = 50 μm. (**b**) Intracellular ROS level was measured by DCF-DA fluorescent probe in cell lysates. n = 3. (**c**) Total antioxidant capacity was measured in cell lysates. n = 3. (**d**) Western blot analysis and densitometric quantification of the protein level of catalase, KEAP1, NRF2, NQO1, SOD1, SOD2, and GAPDH. n = 4. Values are mean ± SEM. * *p* < 0.05, ** *p* < 0.01, and *** *p* < 0.001 matched indicated group. ^#^
*p* < 0.05, and ^###^
*p* < 0.001 matched PA group.

**Figure 4 antioxidants-12-01019-f004:**
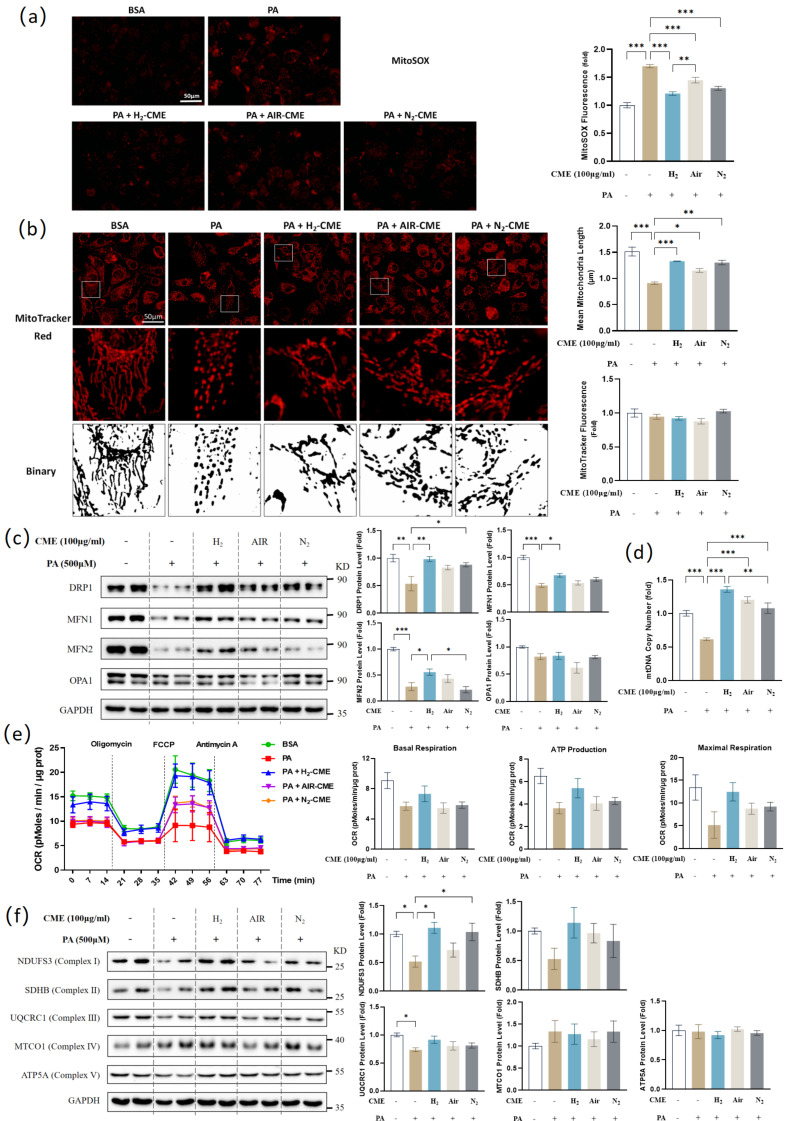
Hydrogen-protected extraction enhanced the capacity of CME to attenuate PA-induced mitochondrial fragmentation and dysfunction in HAEC cells. HAECs were pretreated with 100 μg/mL CME for 24 h, followed by 500 μM PA or BSA treatment for another 24 h. (**a**) Mitochondrial superoxide level was visualized by MitoSOX fluorescent probe and quantified. Magnification: ×40. Scale bar = 50 μm. (**b**) Mitochondrial morphology was visualized by MitoTracker Red fluorescent probe. Binary processing of the images was performed. Fluorescence intensity and mean mitochondrial length were quantified. Original magnification: ×40. Scale bar = 50 μm. (**c**) Western blot analysis and densitometric quantification of the protein levels of DRP1, MFN1, MFN2, OPA1 and GAPDH. n = 4. (**d**) Mitochondrial DNA copy numbers were measured by qPCR. n = 4. (**e**) Oxygen consumption rate (OCR) was measured with Seahorse XF24 Extracellular Flux Analyzers and quantified. n = 4. (**f**) Western blot analysis and densitometric quantification of the protein levels of NDUFS3, SDHB, UQCRC1, MTCO1, ATP5A and GAPDH. n = 4. Values are mean ± SEM, * *p* < 0.05, ** *p* < 0.01, *** *p* < 0.001.

**Figure 5 antioxidants-12-01019-f005:**
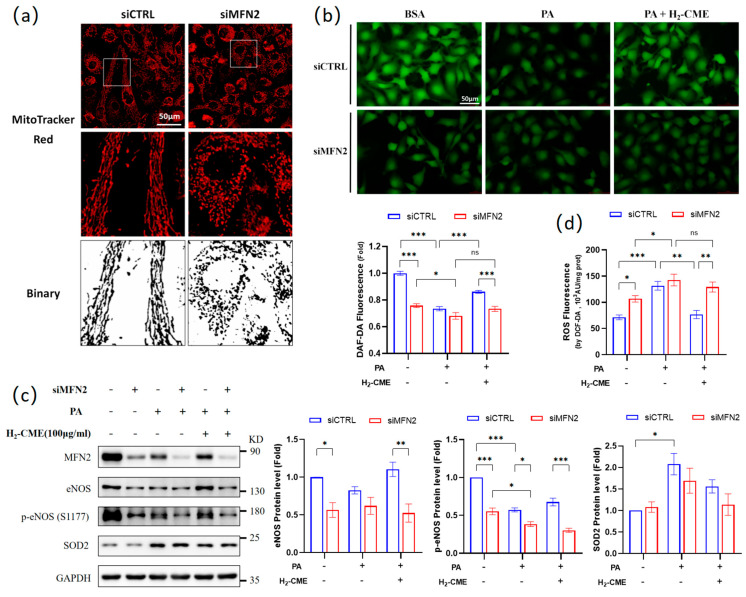
MFN2 mediated H_2_-CME’s prevention of PA-induced decrease of NO production and increase of ROS generation in HAEC cells. HAECs were transfected with siMFN2 or siCTRL, followed by 100 μg/mL H_2_-CME treatment for 24 h, followed by 500 μM PA or BSA treatment for another 24 h. (**a**) Mitochondrial morphology was visualized by MitoTracker Red fluorescent probe. Binary processing of the images was performed. Original magnification: ×40. Scale bar = 50 μm. (**b**) Intracellular NO level was visualized by DAF-FMDA fluorescent probe and quantified. Magnification: ×40. Scale bar = 50 μm. (**c**) Western blot analysis and densitometric quantification of the protein levels of MFN2, eNOS, p-eNOS(S1177), SOD2 and GAPDH. n = 3. (**d**) Intracellular ROS levels were measured by DCF-DA fluorescent probe in cell lysates. n = 4. Values are mean ± SEM, * *p* < 0.05, ** *p* < 0.01, *** *p* < 0.001, ns: not significant.

**Figure 6 antioxidants-12-01019-f006:**
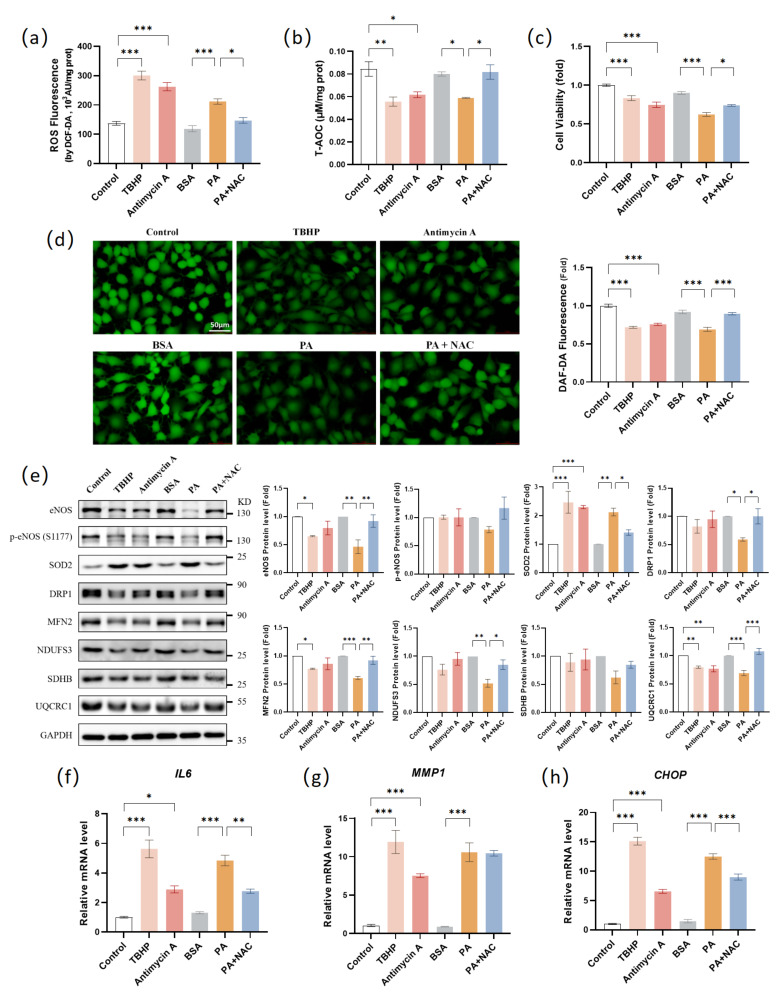
Oxidative stress mediated PA-induced endothelial dysfunction in HAEC cells. HAECs were treated with TBHP (100 μM, 6 h), antimycin A (25 μM, 24 h), BSA (24 h), PA (500 μM, 24 h), and PA (500 μM, 24 h) with NAC (1 mM). (**a**) Intracellular ROS levels were measured by DCF-DA fluorescent probe in cell lysates. n = 3. (**b**) Total antioxidant capacity was measured in cell lysates. n = 3. (**c**) Cell viability was measured by MTT assay. n = 6. (**d**) Intracellular NO level was visualized by DAF-FMDA fluorescent probe and quantified. Magnification: ×40. Scale bar = 50 μm. (**e**) Western blot analysis and densitometric quantification of the protein levels of eNOS, p-eNOS(S1177), SOD2, DRP1, MFN2, NDUFS3, SDHB, UQCRC1, and GAPDH. TBHP and antimycin A values fold change matched control values. PA and PA+NAC values fold change matched BSA values. n = 3. (**f**–**h**) mRNA levels of IL6 (**f**), MMP1 (**g**), and CHOP (**h**) were measured by qPCR. n = 4. Values are mean ± SEM, * *p* < 0.05, ** *p* < 0.01, *** *p* < 0.001.

**Figure 7 antioxidants-12-01019-f007:**
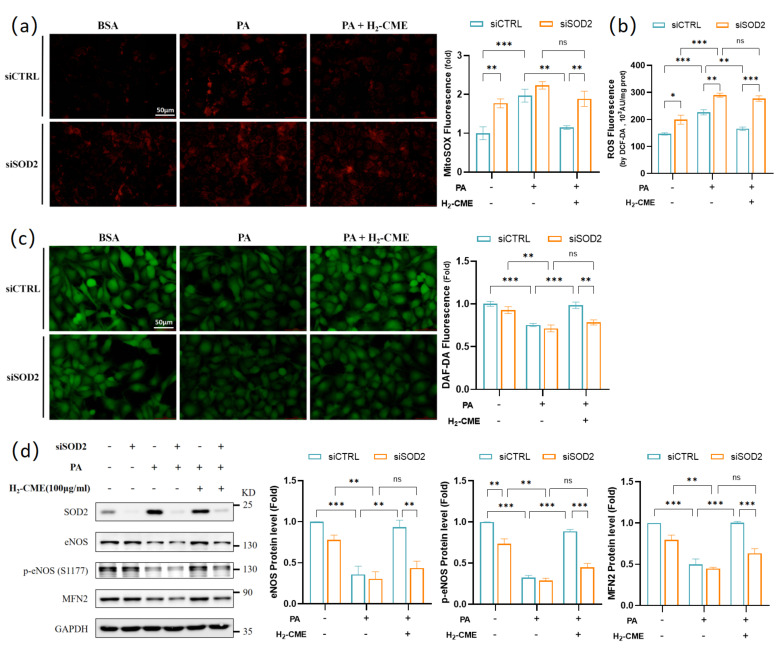
H_2_-CME prevented PA-induced decrease in NO levels by maintaining redox balance in HAEC cells. HAECs were transfected with siSOD2 or siCTRL, followed by 100 μg/mL H_2_-CME treatment for 24 h, followed by 500 μM PA or BSA treatment for another 24 h. (**a**) Mitochondrial superoxide level was visualized by MitoSOX fluorescent probe and quantified. Magnification: ×40. Scale bar = 50 μm. (**b**) Intracellular ROS level was measured by DCF-DA fluorescent probe in cell lysates. n = 3. (**c**) Intracellular NO level was visualized by DAF-FMDA fluorescent probe and quantified. Magnification: ×40. Scale bar = 50 μm. (**d**) Western blot analysis and densitometric quantification of the protein levels of MFN2, eNOS, p-eNOS(S1177), SOD2 and GAPDH. n = 3. Values are mean ± SEM, * *p* < 0.05, ** *p* < 0.01, *** *p* < 0.001, ns: not significant.

## Data Availability

The data presented in this study are available in the article.

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
