# Peer review of "Hydrogen Protection Boosts the Bioactivity of Chrysanthemum morifolium Extract in Preventing Palmitate-Induced Endothelial Dysfunction by Restoring MFN2 and Alleviating Oxidative Stress in HAEC Cells"

_antioxidants, 2023, doi:10.3390/antiox12051019_

Round 1
Reviewer 1 Report
I recommend that the paper be accepted with minor revision:
a) The authors should check for typographical and grammar error the entire manuscript (space, page lines etc…)
b) I recommend that the authors also better mention the objective of in vitro study in the introduction.
c) I recommend that the authors talk about other beneficial effects of CME inserting more references in other in vitro studies or in vivo models.
d) The authors are advised to expand more on the part of oxidative stress in endothelial dysfunction.
e) I recommend that the authors should better explain the function of MFN2
Special comments:
The research analyzed whether Chrysanthemum 2 morifolium's extraction method offered a more effective line of defense against endothelial dysfunction brought on by palmitate.
It is original and interesting and relevant in the field of biochemistry.
The topic is interesting because it demonstrates the importance that new extraction methods are for conserving the phytochemical characteristics of plants and their antioxidant potential.
It should be interesting to evaluate the mechanism of action by in vivo studies. Anyway , the authors should better investigate the role and the mechanism of action by western blots or pcr
The conclusions are indeed consistent.
The authors should ameliorate the quality of figures
The English should be improved
Reviewer 2 Report
This is a very well-conducted study and well-written manuscript in excellent English. It describes an optimised extraction of Chrysanthemum (CME) and addresses the advantages of hydrogen-extracted CME in comparison to the extraction methods for its antioxidant and other beneficial properties in endothelial cells treated with palmitic acid.
I just have a few minor issues that need to be addressed:1. Please explain the abbreviations DPPH and HAEC in the abstract and remove dash between "best" and "prevented" in line 20.
2. Line 21, 330, 364/5, 382, 444, 462, 470, 529, 530, 553, 556 and throughout the ms ( for example lines 330, : Please use the term "expression" only for genes, not for proteins (or RNA) and replace those with "levels/amounts".
3. Lines 35/6: The grammar of the listing is not really correct. Better write "...limited to antioxidant, anti-inflammatory and carcinogenic activities as well as cardiovascular protection."
4. line 39: remove dot after lignans.
5. lines 54, 74 and 501: replace "research" with "studies"
6. line 62: Please specify "homeostatsis"-which one do you refer to?
7. line 64: replace "hightend" with "increased"
8. lines 105 and 115: please explain the abbreviations TPTZ and LC-MS
9. Methods 2.11 and 2.12: Were background/control (auto) fluorescences of unstained cells subtracted from the DHE/DCF-stained cells? This is important as (in particular primary) cells have a certain amount of auto-fluorescence.
10. Line 273: Please correct microm to microM.
11. line 303: better replace "in" with "by" in front of "protecting".
12. lines 332 and 348: There is no need to use capital letters for "catalase", it is not an abbreviation but the name of the enzyme.
13. line 354: remove "the".
14. line 358: "fission and fusion" are nouns, not verbs, so please modify to: "...are constantly UNDERGOING fission an fusion..."
15. line 427: Please combine both sentences with a comma, remove the dot since otherwise grammar is not correct.
16. For heading 3.6 when you say you wanted to assess the role of ox stress in PA-induced EC dysfunction, it sounds as if you used the other stressors in addition to PA which, according to fig. 6 was not the case. Please thus describe the aim of these experiments clearer and state that other stressors were used instead of PA.
17. line 433: Likewise, it is not entirely clear why you used here NAC instead of CME. Please make the rational of using NAC clearer and modify the statement in line 434 since "contribution (of NAC) OF ox stress" does not make much sense, even "TO ox stress" would not be really correct since NAC counteracts ox stress. Please rephrase and clarify.
18. line 498: While you here prominently mention polyphenols, there does not seem any immediate relation to your results unless you describe this connection better.
19. line 501: the term "protection extraction" is not really good English, perhaps replace with "protective extraction.
20. line 515: as above, better say "mitochondria constantly UNDERGO fission and fusion..."
21. line 522: "dynamics (i s singular) WAS disrupted..."
22. line 523: "WAS more severely..."
23. line 524: better say "AMELIORATED... fragmentation" since "improved" could mean more fragmentation...
24. line 526 it should be "effectS"
25. line 534: replace "their" with "it's" since SOD is singular.
The ms is written in excellent English with just very few minor mistakes that I mention to the authors.
Reviewer 3 Report
Hydrogen protection boosts the bioactivity of Chrysanthemum morifolium extract in preventing palmitate-induced endothelial dysfunction by restoring MFN2 and alleviating oxidative stress in HAEC cells by Yilin Gao et al.
The authors present a significant set of results in a logical and well-organized manner, clearly demonstrating that pretreatment with hydrogen prevents oxidative processes during the extraction of Chrysanthemum morifolium components.
The material and methods are adequate, as well as the discussion, so I consider the work acceptable.
Author Response
Thank you very much for your encouraging comments.